# Evaluation of the Properties of Asphalt Mixes Modified with Diatomite and Lignin Fiber: A Review

**DOI:** 10.3390/ma12030400

**Published:** 2019-01-28

**Authors:** Yanchao Yue, Moustafa Abdelsalam, Dong Luo, Ahmed Khater, Josephine Musanyufu, Tangbing Chen

**Affiliations:** School of Human Settlements and Civil Engineering, Xi’an Jiaotong University, Xi’an 710054, China; yuey@xjtu.edu.cn (Y.Y.); luodong@xjtu.edu.cn (D.L.); ahmedkhater@stu.xjtu.edu.cn (A.K.); quinnnatukunda@gmail.com (J.M.); wuhen123@stu.xjtu.edu.cn (T.C.)

**Keywords:** asphalt mix, rutting, thermal cracking, moisture damage, diatomite, lignin fiber

## Abstract

Due to rapid growth of traffic density, the phenomenon of overloading on high-grade highways causes various modes of distresses to the pavement such as rutting, thermal cracking, and water damage. Modification of asphalt mixes is the most common solution to improve the performance of asphalt pavement to mitigate its damages. This paper provides a review on the influence of diatomite or lignin fiber as a modifier in asphalt mixes. In order to assess the effectiveness of selected additives on asphalt mix performance, several tests, such as wheel tracking, indirect tensile, three points bending, freeze thaw splitting, and marshall immersion, were reviewed. The review indicated that the addition of diatomite increases the high temperature rutting resistance of asphalt mixes, but some researchers observed that it has a little improvement on the low temperature performance of asphalt mixes and the optimum amount of diatomite at 12–14% of asphalt binder can be added into the mix. In contrast, lignin fiber has a significant effect on the low temperature cracking resistance of asphalt mixes; however, its influence on the high temperature rutting resistance of asphalt mix is limited, and the optimum amount of lignin fiber is 0.2–0.4% per asphalt mix composition. The review also indicated that the single additives haven’t the ability to enhance the overall performance of asphalt mix. Consequently, the utilization of double additives can improve the overall performance of asphalt mixes at the same time, but it is still in an early stage in the application of highway engineering due to all previous researches concentrated on the single modification. Moreover, this review suggests that the future use of diatomite and lignin fiber compound modified asphalt mix can improve the overall mix performance.

## 1. Introduction

Road networks are mainly classified into two categories: namely, flexible pavements and rigid pavements. Flexible pavements are more vastly used compared to rigid pavement due to advantages such as low initial cost, good resistance to temperature variation, easy repair work, and easy to locate underground works (pipe location) [1]. For example, in China 90% of total pavement structures are flexible pavement [2] while in the U.S., they present about 95% of total pavement structures [3].

Throughout the last decades, there has been a rapid increase in traffic loading intensity in terms of numbers of axles and large tire pressures caused by weighty vehicles. This rapid growth leads to some undesirable distresses in pavements such as rutting (permanent deformation) under the influences of repeated vehicle loading at high temperature, low temperature thermal cracking, and freeze–thaw cycles [4,5,6,7], which decrease the ride quality and service life of road pavements [8]. Therefore, technologists are always trying to enhance the properties of asphalt pavement mixes, where any extension in the service life of road pavements will be of course a great benefit to the economy [9].

Asphalt, or asphalt mixes modified with additives, is the most commonly applied approach to save natural resource and improve pavement performance when the asphalt produced does not meet the weather, traffic loads, and pavement structure requirement [10,11].

There are several types of additives which can be used as a single additive or composite reinforcement in asphalt mix. For more knowledge, most of the literature revealed that a single additive cannot improve all the performances of pavement asphalt mixes at the same time. As an example, asphalt binder with additives like crumb rubber, polymers, and natural rubber have been used to resist the rutting at high temperatures and raveling in asphalt mixes. However, the problem of low temperature cracking still persists. On the other hand, the fibers can improve fatigue life by increasing the resistance to low temperature cracking [12,13]. Thus, the use of double additives may improve the overall performance of asphalt mix, but most of the researchers are focused on the single modification and the composite reinforcement is still in its premature stage [14].

Diatomite and lignin fiber have been selected among the various types of additives. Diatomite is a type of material with high porosity, low density, light weight, high reserves and low cost. Lignin fiber is a kind of fibers with a large surface area, a rough surface and a high heat resistance.

Previously, researchers noted that diatomite has a key role in enhancing the performance of asphalt mixes at high temperature. However, it has a slight influence on low temperature cracking resistance of asphalt mixes. On the contrary, lignin fiber can improve the low temperature performance of asphalt mixes, but some researchers found that the addition of lignin fiber has minor effect on high temperature rutting resistance of asphalt mixes. So it can be verified that single additives do not have the ability to enhance the overall performance of the asphalt mix, hence, using double additives becomes one of the urgent requirements to improve the overall performance of asphalt mix.

The objective of this study is to review the previous research works on the utilization of diatomite [15,16,17,18,19,20,21,22,23,24] and lignin fiber [14,25,26,27,28,29,30] additives in asphalt mixes. More specifically, the proposed study tries also to thoroughly know the future possibility of using diatomite and lignin fiber compound modified asphalt mix to improve the overall performance of asphalt pavement mixes under environmental temperature and water damage effects.

## 2. Pavement Distresses

### 2.1. Rutting

Permanent deformation, basically referring to rutting, is deemed one of the major distresses that affect asphalt pavement performance [31,32,33]; it appears in the form of surface depression in wheel paths along the sides of the pavement [34]. Moreover, water collects in these surface depressions and cannot drain easily off the surface layer, which may cause premature failure, thus rutting is considered hazardous to the safety of pavement. The phenomenon of rutting occurs at intersections, bus stops on urban roads due to the horizontal loading caused by friction between the pavement and tire through applying vehicle braking, and accelerating actions that also create large shear stress and strain in pavement layers. Rutting also may occur in a long and steep section of highways, based on the principle of time-temperature superposition [35,36,37,38]. Rutting behavior is dependent on environmental factors and traffic intensity [39], for instance, repeated heavy vehicle loading and slow traffic at high temperature [40]. Furthermore, design of pavement structure, poor qualities of pavement materials and construction [41,42], classification of highway, and surface age are also important factors for rutting accumulation [43]. Among the various methods applied on asphalt mix to evaluate rutting performance, this review is focused on wheel tracking tests to calculate rutting resistance factor (Dynamic stability).

### 2.2. Low Temperature Thermal Cracking

Thermal cracking is considered one of the main asphalt pavement failures observed in the world [44,45,46]. This mode of pavement distress appears in a series of transverse cracks that usually start on the sides or the center of the pavement structure and propagate across the pavement surface as a result of low temperatures [47]. Thermal cracks appear on the pavement surface in cases when the thermal tensile stress is equal to or exceeding the tensile strength of pavement [48,49]. Thermal cracks occur in a pavement structure due to repeated heating and cooling throughout the day with a big drop in temperature [50,51]. The lower the content of asphalt binder the less effective adhesion is between the aggregate and the asphalt binder and this can reduce the capacity of the pavement structure to bear the stress. Not only material and weather factors, but also the pavement design (thickness of layers and width of highway) contribute to low temperature thermal cracking [52]. The critical problem concerning this distress when the water penetrates the asphalt layers and fills these cracks [53] is it freezes, and frost heave may increase, which leads to premature deterioration of the pavement structure [54] due to the reduction in adhesive between the aggregate and the asphalt binder. Moreover, thermal cracking may cause various problems such as reduced service life, poor ride quality, and high maintenance costs [55]. Recently, many methods are available to assess the low temperature cracking resistance of asphalt mix; in this study, indirect tensile and three point bending tests are reviewed.

### 2.3. Moisture Damage

Water damage has been a key problem for asphalt pavement engineers for many years [56], which is defined as loss of the adhesive bond between asphalt binder and aggregate due to the presence of water [57], and it can be estimated depending on the loss of mechanical properties in asphalt mixes. In asphalt mixes, asphalt serves two major purposes: to grip the aggregates firmly and to act as a sealant against water entry [58]. Asphalt mix is very sensitive to the existence of humidity on the surface layer of pavement [59]. The presence of water is inevitable, where it can exist in the pavement from many sources such as flow of rain, snow, and capillary absorption of ground water into the pavement layers [60]. Water can get into the pavement structure in various ways, through the cracks on the surface or interconnectivity of the air voids from the road shoulders, or rising in the level of ground water, as well as through the high porosity in asphalt mix that can lead to the ingress of damaging water and air [61]. The penetration of water into the pavement causes most of the distresses like raveling, rutting, fatigue damage, and stripping [62,63]. The resistance of asphalt pavement to moisture damage is impacted by numerous factors such as type of aggregate, properties of asphalt mixes, grade of bitumen, additives, pavement structural design, environmental conditions, and traffic intensity [64]. Several methods have been advanced for estimating the water stability of asphalt mix such as Boiling Water Test, Freeze-Thaw Test, Marshall Immersion Test, SHRP Moisture Susceptibility Study, Net Adsorption Test (NAT); however, this study is concentrated on the most used tests freeze-thaw splitting and the Marshall Immersion Test.

## 3. Preparation of Sample and Laboratory Test Methods for Asphalt Mix

### 3.1. Preparation of Sample Modified with Additives

There are two methods available for the mixing of the additives in asphalt mixes: the dry process and the wet process. 

Generally, the dry process is the most applied method with fibers. First, lignin fibers are mixed with the hot aggregates for a minimum of 30 s to enhance fiber dispersion. Secondly, the bitumen is added while continuing to mix for nearly 2 additional min. The overall processing time should not exceed 6 min in order to prevent asphalt aging.

Diatomite can be mixed into asphalt by using the wet process, where diatomite and bitumen are weighed to the required amount and left in the oven at 135 °C for 4 h. The diatomite and bitumen are then taken out of the oven and mixed together with the blending equipment at a speed of 600 r/min for 15 min.

### 3.2. Evaluation of High Temperature Performance

#### 3.2.1. Wheel Loading Tracking Test (Rutting Test)

The Wheel loading tracking test is an important tool to assess the high temperature rutting resistance of various asphalt mixes in the laboratory [65], where it is easy to understand and apply. According to standard specification [66] and ASTM WK64214, three slab samples (300 mm length × 300 mm width × 50 mm high) were prepared and mixed under the O.A.C and compacted with a hammer 24 times. Initially the test samples were placed in an atmosphere at 60 ± 0.5 °C for 6 h. After this, the plates are loaded with a pressure of a single rubber wheel with a tire pressure of 0.7 MPa for 60 min within a traveling distance of 230 ± 10 mm and a speed of 42 ± 1 cycles/min. The wheel loading tracking test can also be carried out at different temperatures from (50–80) °C. Throughout the test, LVDTs are used to calculate the vertical displacement of the slab sample and deformation (rutting) depth was recorded every 20 s [67]. Dynamic Stability (DS) is determined to evaluate the rutting performance of asphalt mixes [68,69] by applying the following equation:(1)$$\mathrm{DS}=\frac{t_{2}-t_{1}}{d_{2}-d_{1}}\times C_{1}\times C_{2}\times N=\frac{15\times 42}{d_{2}-d_{1}}$$
where: d_1_ is the rutting depth (mm) at (t_1_ = 45 min); d_2_ is the rutting depth (mm) at (t_2_ = 60 min); C_1_, C_2_ are the correction factors of equipment and samples, respectively, (C_1_, C_2_ = 1) for this equipment; N is the value of the running speed test wheel (42 cycles/min).

### 3.3. Evaluation of Low Temperature Performance

#### 3.3.1. Indirect Tensile Test (Splitting Test)

The Indirect tensile (IDT) test was advanced in the USA during the Strategic Highway Research Program (SHRP) to evaluate asphalt mix performance in cold regions [70,71]. The splitting test is applied on three identical cylinder samples with dimensions of 101.6 mm in diameter and 63.5 mm in height at a temperature of −10 °C with a loading rate of 1 mm/min according to Chinese standard specifications [66] and ASTMD6931. Before the test all samples were compacted 75 times on each side using a hammer, then stored inside a chamber at the temperature of the test −10 °C for 3 h. Once the test began, the samples were removed from the chamber and employed in a short time to avoid any change in the temperature of samples. During the test, indirect tensile (split) strength and tensile strain can be calculated [66] to reveal the properties of asphalt mixes at low temperature by using the following Equations.
(2)$$R_{t}=\frac{0.006287P_{t}}{h}$$
(3)$${Ɛ}_{t}=\frac{X_{t}\left(0.0307+0.0936µ\right)}{\left(1.35+5µ\right)}$$
(4)$$X_{t}=\frac{Y_{t}\times \left(1.35+0.5µ\right)}{1.794-0.0314µ}$$
where: R_t_ is the split strength (MPa); P_t_ is the tensile failure load (N); h is the height of the sample (mm); Ɛ_t_ is the tensile strain (N); X_t_ is the horizontal deformation (mm); Y_t_ is the vertical deformation (mm); µ is the Poisson’s ratio; it is 0.25 in this test.

#### 3.3.2. Low Temperature Bending Test (Three Point Bending Test)

The Low Temperature Bending Test is carried out to evaluate tensile strength or resistance to thermal cracking, which may be occurring in asphalt mixes under the effects of low temperatures [6,45]. Before the test, four duplicate beam samples with size 250 mm (length) by 30 mm (width) by 35 mm (height) are fabricated and submerged in a constant temperature container at −10 °C for 3 h. After this, the three points bending loading method is employed on a span length of 200 mm at −10 °C temperature and 50 mm/min loading rate, following the specification [66] and AASHTO 321, by using the Material Testing System (MTS-810). The maximum flexural stress and the maximum tensile strain are determined as follows [68]:(5)$$R_{B}=\frac{3{\mathrm{LP}}_{B}}{2{\mathrm{bh}}^{2}}$$
(6)$${Ɛ}_{B}=\frac{6\mathrm{hd}}{L^{2}}$$
where R_B_ is a flexural strength (MPa); Ɛ_B_ is tensile strain (N); P_B_ means the load when failure occurs (N); d is the deflection when failure occurs (mm); h is the height of the beam (mm); b is the width of the beam (mm); L is the span length of the beam (mm)

### 3.4. Evaluation of Water Stability Performance

#### 3.4.1. Freeze-Thaw Splitting Test

The Freeze-thaw splitting test (FTST) is conducted to estimate the water stability of asphalt mix through the comparison of the indirect tensile strength ratio before and after the moisture damage [72]. Eight duplicate cylindrical samples with a size of 101.6 mm (diameter) by 63.5 mm (height) are fabricated and compacted 50 times on both sides using a hammer and divided into two groups, each group consisting of four samples [66]. The first group is cured in water at 25 °C for 20 min. The second group is saturated by using the vacuum for nearly 15 min under a pressure of 98.3–98.7 kPa. After saturation, the samples are sealed in plastic leak proof bags containing 10 mL of water. The wrapped samples are stored in a freezer at a temperature of −18 °C for 16 h and then placed in a warm water bath at 60 °C for 2 h. The two groups (Frozen thawed and fresh samples) are bathed in water at 25 °C for more than 2 h. Finally, the indirect tensile strength of the samples is performed at 25 °C with the loading rate of 50 mm/min, following the specification AASHTO T283 [73], after which tensile strength ratio (TSR) is obtained as the following equation [69]:(7)$$R_{t}=\frac{0.006287P_{t}}{h}$$
(8)$$\mathrm{TSR}\%=\frac{R_{t2}}{R_{t1}}\times 100$$
where TSR is the tensile strength ratio; Rt_2_ is the Splitting strength of frozen thawed samples (MPa); Rt_1_ is the Splitting strength of fresh samples (MPa).

#### 3.4.2. Marshall Immersion Test

The immersion–compression test was presented as the first test to assess the water damage on compacted asphalt mix samples in 1950 [74]. Two equivalent groups of Marshall Test samples were prepared and compacted 50 times using a hammer on both sides, each group consisting of three identical samples with sizes 101.6 mm in diameter by 63.5 mm in height. One group was submerged in water at 60 °C for 30 min, while the other group of the samples was submerged in a warm water at a 60 °C constant temperature for 48 h according to the standard specification [66] and ASTM D1559. Thus, the immersion residual Marshall stability (MS) was calculated [68] by applying the following equation.
(9)$$\mathrm{MS}\%=\frac{{\mathrm{MS}}_{2}}{{\mathrm{MS}}_{1}}\times 100$$
where: MS is residual stability; MS_2_ is Marshall Stability at 60 °C after 48 h water immersion; MS_1_ is Marshall Stability at 60 °C after 30 min water immersion.

## 4. Influence of Selected Additives on Asphalt Mix Performance

### 4.1. Diatomite

#### 4.1.1. Properties of Diatomite

Diatomite, or Diatomaceous earth, is a natural siliceous sedimentary rock crushed to powder with a white or off-white in color, size from less than 1 μm to more than 1 mm, but is normally from 10 to 200 μm [75]. It is made of the remains of fossilized skeletal single-cell aquatic plants called diatoms [76]. At the end of 1996, Chinese reserve of diatomite resources exceeded 300 million tons [77], ranking second in the world after the United States [78]. Diatomite has a light weight, large surface area, low cost, wear-resistant, low density, a high porosity, strong adsorption [78,79], sound insulation, heat conduction, low thermal conductivity, corrosion resistant, non-polluting, non-toxic characteristics [78], and hasn’t any chemical reaction with asphalt [80]. Diatomite also has a high melting point ranging from 1000 °C to 1750 °C [81,82], with a bulk density of 0.35–0.42 g/cm^3^, specific gravity of 2.1–2.3 g/cm^3^, and pH of 7. Due to the aforementioned characteristics, diatomite has gained great interest in China where it has been widely used for asphalt mix with the purpose of reducing environmental pollution, decreasing the consumption of natural resources and improving asphalt pavement mix performance [83]. For more information on the image of diatomite and its microstructure, refer to [15,16].

#### 4.1.2. Effect of Diatomite on Asphalt Mix Performance

Until now, research in the field of asphalt mixes modified with diatomite is still continuing, and most of the studies are summarized in Table 1 and Table 2. Table 1 refers to asphalt type mix, bitumen type and amount, aggregate and filler types, and gradation of asphalt mix. Table 2 refers to the type and amount of diatomite modifier and the results of high temperature performance, low temperature performance, and water stability.

Test procedures discussed in chapter three have been applied by all previous research in terms of sample preparation, storage, and the number of tested samples. The following paragraphs review previous research on the effect of diatomite in asphalt mixes, whose components are included in Table 1 with results in Table 2.

Diatomite was used in a 2018 study by Chao Yang et al. [15] for evaluating and improving the properties of asphalt mix. A three point bending test was carried out and its result detected that diatomite has a slight influence on the improvement of the low temperature stability of an asphalt mix. Contrarily, the wheel tracking test observed that asphalt mix modified with diatomite improved the dynamic stability (DS) value by 3.4 times compared to the base asphalt mix. Freeze–thaw splitting and the Marshall Immersion Test showed that with the addition of diatomite, the moisture susceptibility values of modified asphalt mix are greater than that of base asphalt mix.

Cheng et al. [16] presented a study on the low temperature performance of sand asphalt modified by varying the ratios of diatomite to lime stone. According to the results of the splitting test, the low temperature cracking resistance of asphalt mixes could be improved by the addition of diatomite, where with the addition by 50% of lime stone powder (8.1% of lime stone powder and 6.5% of diatomite) of total mass of aggregate, the splitting strength is increased by 1.37 times compared to the net asphalt mix. However, with the increase of diatomite content after 50% of lime stone powder, splitting strength values decrease, but remain better than the net asphalt mix.

Davar et al. [17] conducted research on diatomite and basalt fibers compound on improved tensile failure strength of asphalt mix at low temperatures. An indirect tensile test indicated that asphalt mix modified with diatomite has a bad tensile failure strain behavior compared with the ordinary sample, where it recorded the lowest tensile failure strain value between all samples. However, diatomite and basalt fibers compound modified asphalt mix obtained a greater tensile failure strain and tensile strength by 13% and 22%, respectively, than the ordinary sample.

Athma et al. [18] investigated the influence of diatomite on the performance of asphalt mix using a water stability test. Results reported that diatomite modified asphalt mix enhances water damage resistance where the TSR value of diatomite modified asphalt mix is 91.9%, and this value is higher than 80% (minimum requirement of TSR value).

Luo et al. [19] used various contents of diatomite modified asphalt mixes in a study on pavement performance. A high temperature rutting test on different content of diatomite demonstrated that DS values of the diatomite modified asphalt mixes is better than the plain asphalt mix at about 2:2.8 times. Moreover, the freeze-thaw splitting test revealed that asphalt mixes containing various quantities of diatomite obtained a higher TSR by 9.41%:12.96% than plain asphalt mix.

Guo et al. [20] introduced Diatomite and Glass fiber Compound Modified into the Asphalt Mix (DGFMAM) and undertook wheel loading tracking and splitting tests on asphalt mix reinforced. Rutting results detected that the maximum DS of DGFMAM increased by 2 times of the corresponding value of the unmodified asphalt mix. Splitting test results detected that tensile strength of DGFMAM experiences no clear change at low temperatures. In contrast, DGFMAM can improve the tensile failure strain; it was also noted that glass fiber plays a key role in this enhancement.

The effect of diatomite and polyethylene (PE) particles on low temperature properties of asphalt mix was examined by Hu and Zhou [21]. A flexural test recorded that diatomite has a small degree of improvement on low temperature performance of asphalt mix. Nevertheless, the utilization of diatomite and PE composite modified asphalt mix improves the low temperature crack resistance by 42.8% compared to the net asphalt mix. This indicates that diatomite has a slight effect on the low temperature properties of asphalt mixes, but the PE has a significant influence on the low temperature crack resistance.

The rutting test was conducted by Zhao and Li [22] to evaluate the permanent deformation performance of diatomite modified asphalt mix. Dynamic stability values demonstrated that diatomite modified asphalt mix is higher than a common asphalt mix by about 2.2 times.

Zhu et al. [23] published the result of a study in which they had assessed the effect of diatomite modified asphalt mix in permafrost regions. The tests included wheel loading tracking and freeze-thaw cracking. Results denoted that the DS, residual stability, and TSR values of asphalt mixes modified with diatomite are enhanced by 55.3%, 2.4% and 9.6%, respectively, compared to the control asphalt mix.

Laboratory investigations were carried out by Tan et al. [24] on the low temperature performance of asphalt paving mixes. Bending test results observed that diatomite modified asphalt mix is worse than the plain asphalt mix about 9.5%, 6.6% in tensile stress and tensile strain, respectively.

From the previous studies, it is verified that diatomite can improve the high temperature performance of asphalt mixes due to its large surface area and porous structure, and can absorb slight contents of bitumen, which increases the all complex shear modulus of the asphalt and improves rutting resistance of the mix.

### 4.2. Lignin Fiber

#### 4.2.1. Properties of Lignin Fiber

Lignin is one of the most abundant renewable organic resources in the world [84]. Lignin is an antioxidant material that can resist oxidation in asphalt binder, which is considered the main reason for long-term aging in asphalt pavements [85]. Lignin fiber was used as an additive to improve the performance of the asphalt mix, where its rough surface, beneficial to the adhesion of bitumen, and its large surface area enables absorption and stabilization of bitumen, increasing asphalt binder content and enhancing crack resistance [4]. Moreover, lignin fiber has a high heat resistance, so it can be combined with the asphalt mix at a high temperature, where its melting point is more than 200 °C, specific surface area is 118.1 (10^−3^ m^2^/g) and density is 1.28 g/cm^3^. For more information on the image of lignin fiber and its microstructure, refer to [28].

#### 4.2.2. Effect of Lignin Fiber on Asphalt Mixes Performance

The majority of researches undertake on the influence of lignin fiber into asphalt mixes were summarized in Table 3 and Table 4. Table 3 refers to asphalt type of mix, bitumen type and amount, aggregate and filler types and gradation of asphalt mix. Table 4 refers to the type and amount of lignin fiber modifier and the results of high temperature performance, low temperature performance, and water stability.

Test procedures discussed in chapter three have been applied by all previous research in terms of sample preparation, storage, and the number of tested samples. The following paragraphs review previous research on the effect of diatomite in asphalt mixes, whose components are included in Table 3 with results in Table 4.

Fu et al. [14] conducted another investigation on asphalt mix modified with Composite Reinforcing Material (CRM). The high temperature rutting test denoted that CRM can improve the permanent deformation behavior of asphalt mix, as well as the DS values of asphalt mixes modified with CRM are higher than the control sample. According to the result of the bending test, the low temperature cracking resistances of asphalt mixes are enhanced by adding CRM compared with the control sample; moreover, it is remarked that the flexural tensile strength and tensile strain reaches the maximum value by adding 0.4% and 0.8%, of CRM, respectively. Residual stability and TSR values showed improvement in water stability of asphalt mix by adding 0.4% of CRM. Nevertheless, with the increase of CRM content after 0.4%, TSR and residual stabilities values started to decrease but were still better than the control sample.

Recently, a comparative study carried out by Fu et al. [25] on two different modified asphalt mixes, one modified with double additives (anti-rutting agent and lignin fiber) and the other modified with only lignin fiber. Wheel load tracking, indirect tensile strength, and water stability occurred. Results reported that DS values of AC-16 and AC-13 asphalt mixes modified with double additives are improved by 6.8 and 7.3 times, respectively, compared to asphalt mixes modified with lignin fiber only. Failure strain results revealed that lignin fiber plays a crucial role in enhancing the performance of asphalt mixes at low temperatures; besides, it has nearly the same effect on asphalt mix performance at low temperature compared with double additives. Results of moisture susceptibility of AC-16 and AC-13 asphalt mixes modified with double additives are enhanced by (3.8% and 6.4%, respectively, for Residual stability) and by (3.8% and 6.4%, respectively, for TSR) compared with lignin fibers modified asphalt mixes.

Based on extensive research carried out by Rui et al. [26], asphalt mix samples with lignin fibers were manufactured at 4.95% optimum asphalt content. A wheel tracking test remarked that rutting resistance of lignin fiber modified asphalt mix is slightly improved. The three point bending test indicated that the low temperature stability of asphalt mix is increased with the addition of lignin fiber. The freeze–thaw splitting test observed that lignin fiber reinforced asphalt concrete obtained a better TSR than the unmodified asphalt mix.

Tuya [27] presented a comparative study on two various modified asphalt mixes. The dynamic stability of the wheel tracking test revealed that lignin fiber and rubber powder compound modified asphalt mix is better than an asphalt mix modified with rubber only. The three point bending test results showed that the low temperature cracking resistance of asphalt mix modified with lignin fiber and rubber powder is enhanced compared with asphalt mix modified with rubber only. Water stability results recorded improvement in the moisture susceptibility of asphalt mixes modified by 0.2%, 0.25% of lignin fiber compared with an asphalt mix modified with rubber only. However, by adding 0.3% of lignin fiber the TSR and residual stability values decreased, in addition to these values not meeting the minimum requirements of specification greater than 80%.

Xu et al. [28] surveyed the effects of lignin Fiber on the performance of asphalt mix under water effects and environmental temperature. The high temperature rutting test noted that lignin fiber reduces the asphalt mix rutting depths compared with ordinary mix. A low-temperature flexural test was employed and observed that the low temperature behavior of asphalt mix is increased by adding lignin fiber compared with ordinary mix. The freezing–thaw cycling test indicated that TSR of lignin fiber modified asphalt mix had a bad performance in resisting water damages compared with ordinary mix.

Van et al. [29] used various lignin fiber contents in a study of Stone Mastic Asphalt mixes (SMA), which was assessed by wheel loading and water stability tests. Not only dynamic stability but also residual stability ratio values indicated that lignin fiber improves the performance of asphalt mix by adding 0.3%, but if the content of fibers continues to increase, the DS and residual stability values are decreased slowly.

Wheel loading tracking and moisture susceptibility test were conducted by Xu et al. [30] to analyze the performance of three various types of fibers (lignin fiber, mineral fiber, and cotton fiber) on enhanced SMA. Results demonstrated that 0.3% of lignin fiber has little effect on the resistance of rutting at high temperature compared with the other types of fibers, but it has a great influence on the resistance of water damages in asphalt mix.

From the above mentioned studies, it can be concluded that the adding of lignin fiber improves the asphalt mix properties and increases the absorption of bitumen content, which is beneficial to the anti-cracking at low temperature. Moreover, the toughening effect of lignin fiber is derived from the residual stress and strain fields and the micro cracks around the fiber matrix interface result from the different material properties between fiber and mixes.

## 5. Conclusions and Recommendation

This paper intends to review the utilization of diatomite or lignin fiber in asphalt mixes modification to improve the performance of asphalt mixes and decrease pavement damage like rutting, thermal cracking, and water damage. In addition, many tests that were carried out to evaluate the efficiency of diatomite or lignin fiber on improvement of high temperature, low temperature, and water stability performance of asphalt mixes, were reviewed. Based on the above analysis presented herein, conclusions and recommendations of this review are summarized as follows:Diatomite significantly enhances the high temperature performance of asphalt mixes, though some of the research detected that the improvement of low temperature performance of asphalt mixes was insignificant, while others recorded that it has bad behavior on the resistance of low temperature cracking.The low-temperature crack resistance of asphalt mixes is increased by the addition of lignin fiber, and slightly improves the high temperature rutting resistance of asphalt mixes.Diatomite and lignin fiber have an important effect on water damage resistance in asphalt mixes.According to previous studies, it is noted that the optimum amount of diatomite is 12–14% of asphalt binder can be added into mix and the optimum amount of lignin fiber is 0.2–0.4% per asphalt mix composition.Asphalt mixes modified with single additives cannot enhance the overall properties of asphalt mixes.Double-adding technology in asphalt mixes is the best alternative to decrease low temperature cracking and rutting at high temperatures at the same time.It is recommended that asphalt mixes modified with two varieties of additives (diatomite and lignin fiber) can be utilized in future study to improve the overall performance of asphalt mixes under environmental conditions.

## Figures and Tables

**Table 1 materials-12-00400-t001:** Component of asphalt mixes modified with diatomite [15,16,17,18,19,20,21,22,23,24].

Ref.	Asphalt Mix Type	Bitumen Type	O.A.C *%	Aggregate Type	Filler Type	Gradation of Asphalt Mix
[15]	Coarse asphalt	Shell 70	N/A	Basalt	Limestone	AC-13
[16]	Sand asphalt	Shell 70	N/A	Basalt	Limestone	AC-13
[17]	Sand asphalt	PG64-22	5.80	Rocky mountainous	N/A **	Nominal maximum aggregate size of 19 mm
[18]	Coarse asphalt	PG76	5.25	N/A	N/A	Nominal maximum aggregate size of 14 mm
[19]	Coarse asphalt	Shell 70	4.20	limestone	N/A	AC-20
[20]	Coarse asphalt	Shell 90	N/A	limestone	N/A	AC-13
[21]	Coarse asphalt	Shell 90	4.00	limestone	N/A	AC 25
[22]	Coarse asphalt	Shell 90	4.20	N/A	N/A	AC-20
[23]	Coarse asphalt	Shell 110	N/A	Granite	Limestone	AC-13
[24]	Coarse asphalt	Shell 90	4.50	N/A	Limestone	AC-20

* Optimum Asphalt Content; ** Not Available.

**Table 2 materials-12-00400-t002:** Part of summary of diatomite modified asphalt mix for improving pavement performance [15,16,17,18,19,20,21,22,23,24].

Ref.	Modifier		High Temperature Performance (60 °C)	Low Temperature Performance (−10 °C)		Water Stability Performance	
	Modifier Type	Modifier Content (% of Asphalt Binder)	Dynamic Stability (time/mm)	Max. Tensile Stress (MPa)	Max. Tensile Strain (µƐ)	Residual Stability (%)	TSR (%)
[15]	Diatomite	0	1645	7.91	1130.84	83	85
		12	5625	8.14	1352.72	88	94
[16]	Diatomite + limestone powder	0 + 16.2*	----------	1.77	----------	----------	---------
		3.2* + 12.2*		2.08			
		6.5* + 8.1*		2.35			
		9.4* + 4.1*		2.25			
		13* + 0		1.95			
[17]	Diatomite + Basalt fiber	0.0	----------	1.68 at (−5 °C)	8350 at (−5 °C)	----------	---------
		15 + 0.0		1.96 at (−5 °C)	7390 at (−5 °C)		
		15 + 0.3 **		2.05 At (−5 °C)	9430 At (−5 °C)		
[18]	Diatomite	2	----------	----------	----------	----------	91.9
[19]	Diatomite	0	1027	----------	----------	----------	77.35
		9	2075				86.76
		11	2496				87.88
		13	2865				90.31
		15	2633				87.85
[20]	Diatomite + Glass fiber	0.0 + 0.2 **	696	3.42	3000	----------	---------
		0.1 **+ 0.2 **	919	3.8	3175		
		0.2 **+ 0.2 **	1088	3.55	3450		
		0.3 ** + 0.2 **	1260	3.4	3300		
[21]	Diatomite + PE	0	----------	8.4	887.8	----------	---------
		14 + 0		9.7	978.2		
		14 + PE		12	1248.6		
[22]	Diatomite	0	1186	----------	----------	----------	---------
		6	2517				
[23]	Diatomite	0	1321	----------	----------	95.7	89.4
		20	2051			98	98.6
[24]	Diatomite	0	----------	7.19	2000	----------	---------
		14		6.57	1875		

* Modifier content of the total mass of aggregate; ** Modifier content percent of the asphalt mixture.

**Table 3 materials-12-00400-t003:** Component of asphalt mixes modified with lignin fiber [14,25,26,27,28,29,30].

Ref.	Asphalt Mix Type	Bitumen Type	O.A.C *%	Aggregate Type	Filler Type	Gradation of Asphalt Mix
[14]	Coarse asphalt	Shell 70	4.3 (0.00% CRM) 4.6 (0.40% CRM) 5.0 (0.80% CRM) 5.4 (1.20% CRM)	Limestone	Limestone	AC-13
[25]	Coarse asphalt	Shell 70	4.5 (AC-13) 4.3 (AC-16)	Limestone	Limestone	AC-13 & AC-16
[26]	Coarse asphalt	Shell 90	N/A **	Basalt	N/A	AC-16
[27]	SMA	Shell 90	4.0	Basalt	Limestone	AC-13
[28]	Coarse asphalt	Shell 90	5.6	N/A	N/A	AC-13
[29]	SMA	Shell 70	5.8 (0.26% L.F) 6.2 (0.30% L.F) 6.6 (0.34% L.F) 7.0 (0.38% L.F)	Limestone	Limestone	AC-16
[30]	SMA	Shell 90	6.0	Limestone	Limestone	AC-16

* Optimum Asphalt Content; ** Not Available.

**Table 4 materials-12-00400-t004:** Part of summary of lignin fiber modified asphalt mix for improving pavement performance [14,25,26,27,28,29,30].

Ref.	Modifier		High Temperature Performance (60 °C)	Low Temperature Performance (−10 °C)		Water Stability Performance	
	Modifier Type	Modifier Content (% of Asphalt Mixes)	Dynamic Stability (time/mm)	Max. Tensile Stress (MPa)	Max. Tensile Strain (µƐ)	TSR (%)	Residual Stability (%)
[14]	CRM (62% polymers + 38% Lignin Fibers)	0.0	----------	----------	----------	80.8	88.2
		0.4	↑3.0 times	↑34.6%	↑34.40%	91.5	93.9
		0.8	↑5.6 times	↑32.43%	↑38.10%	89.7	90.3
		1.2	↑8.0 times	↑30.25%	↑32.49%	83.7	89.3
[25]	Only Lignin Fiber (AC 16, AC 13)	0.0	----------	----------	----------	82.9, 82.5	88.13, 87
		0.36	Effect is not obvious	----------	↑1.98, 2.04 times	84, 83.7	88.5, 89.7
	Lignin fiber + Anti-rutting agent (AC 16, AC 13)	0.36 + 0.40	↑7.9, 8.4 times	----------	↑2.00, 2.10 times	93, 90.7	91.9, 95.45
[26]	Lignin Fiber	0.0	----------	----------	----------	77.5	----------
		0.3	↑11.1%	↑12.2%	↑9.00%	80.4	----------
[27]	Lignin Fiber + Rubber powder	0.0 + 0.13	4272	11.17	2616.02	73.67	82.31
		0.2 + 0.13	4634	13.77	3105.42	82.56	88.71
		0.25 + 0.13	5112	14.06	3264.70	88.00	90.35
		0.3 + 0.13	5480	14.17	3473.47	77.74	80.16
[28]	Lignin Fiber	0.0	----------	----------	----------	78.8	----------
		0.3	Rutting depth ↓8.4%	↑11.8%	↑6%	68.1	----------
[29]	Lignin Fiber (L.F)	0.26	5875	----------	----------	----------	84.73
		0.30	7172				89.53
		0.34	6554				88.05
		0.38	5249				85.36
[30]	Lignin Fiber	0.3	5877	----------	----------	63.6	93.1

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
