# Peer review of "Evaluation of the Properties of Asphalt Mixes Modified with Diatomite and Lignin Fiber: A Review"

_materials, 2019, doi:10.3390/ma12030400_

Round 1
Reviewer 1 Report
Dear Authors,
Thank you for your prepared manuscript. It is difficult to write reviews and requires certain efforts. You have done a good work, but some more efforts are required.
My first critical comment will be about the list of references you have provided: you have 108 references (which is quite an amount) but only part of them refers to the application of lignin fiber and diatomite in the asphalt mix / pavement. Why? I don’t understand why you need to mention references with epoxy resin, gneiss, slag etc. Your main focus is at application of lignin and diatomite, so? You may decrease number of references and select only those which are really significant for the subject of your paper. But for example, to have 11 references in the paragraph for describing what is lignin is a bit too much too.
Another critical comment is about the description of test methods, extraction of info from standard, why? usually experts reading your paper will need just a reference to standard mentioned in your paper. Therefore, I suggest to rewrite chapter 3.
Actually, I would suggest to reorganize your paper: start with short info on the problems of distresses in the flexible pavements in your intro; then you provide short info on the materials – diatomite and lignin fiber; then you write critical review about application of those additives in asphalt mixes with sub-chapters mentioned in chapters 2 and 3, in that way chapter 4 will be fully covered. Also, you need to write recommendations to the readers, preferably if you have practical experience mixing those additives in asphalt mix and your knowledge is not only based on the publications you have read. Also to provide comparison with other additives (here you may use the references about other materials but not in test description part as you did previously).
More details have to be added into text, in order when reading it would be clear: which mixes used, conditions, amounts of additives, obtained results and comparison of those results. Technological aspects are mentioned in the references you refer to? It is important, because asphalt material is quite sensitive to the additives and they immediately affect its compaction, etc. There are a lot of studies, but all of those studies have a practical application? Reading your paper I need to see what is the optimal amount of addition of lignin and diatomite, for which mixes, for which environment/region and maybe you may provide asphalt mix design those additives are used? In case I would like to try lignin or diatomite for production of asphalt mix, I prefer to read a study where I can find already all basic necessary information. I miss it in your review.
In your conclusions it should be stated why lignin and diatomite should be used in asphalt mixes, what are advantages and disadvantages, optimal amount of those materials could be added into mix, what type of mix, etc.
1. 4 Line: Josephine Musanyufu?(name, surname) Also you don’t need to add “1” next to names if all authors are from the same institution.
2. 10-11 Lines: I would suggest rephrasing this sentence, it can be written in a shorter form, basically the idea is that traffic density and overloading is drastically increased in comparison to some years ago. Percentage?
3. 11 Line: Pavements “suffer”, please insert another term.
4. 14 Line: impairements?
5. 18 Line: The review demonstrated? Indicated?
6. 19,57,62 Lines: replace “scholars” with “researchers”
7. You need to mention in your abstract the percentages of optimal fiber amount could be applied in the asphalt mixtures, and if you mention high or low temperatures then you need also show the range. Low might be 0 or -30, high +30 or +70.
8. 32-34 Lines: Experts reading your paper do not need definition of the flexible pavements, therefore, I would suggest to remove those lines.
9. 34-36 Lines: the percentage of flexible pavements applied worldwide? Main advantage in comparison to concrete pavements?
10. 37-39 Lines: swift? Where this sentence comes from? I suggest to rewrite it.
11. 44 Line: please choose which term you use throughout your manuscript “mix” or “mixture”. I would suggest “mix”. But don’t use both.
12. 44-45 Lines: Increase in thickness of layer is not really a good solution, if you have crack at the bottom the thickness of layer doesn’t really matter at all. Please add reference straight away in this sentence. I also suggest to add several references from your list who mention that increase of layer thickness is a solution.
13. 44-49 Lines: only two solutions?
14. 58 Line: Please add reference
15. 68 Line: water effects? Specify please
16. 84 Line: this study is focused on wheel tracking test… study or review?
17. 87 Line: I don’t really agree that cracking failures observed in the cold areas of the world. It is observed everywhere. Please indicate what type of cracking you mention in this line. Combine it with lines 92-93.
18. 88 Line: manifested? Another term please.
19. 119 Line: define/list which methods?
20. 121 Line: change to “freeze-thaw splitting”
21. 128-132 Lines: Strange switch, I think you need to start sentence not with “before the test” but in “According to standard [73]…”
22. I would suggest to rewrite/add more info 3.1.1. it doesn’t contain sufficient info for the review. Please in all your references in the list of references, mention all names of all co-authors to the publications you refer. It can be mentioned in the text, for example, Moreno et al. but in list of references the reference should contain all names contributed for that particular study.
23. 130 Line: 60 degrees? The temperature of the mix? What type of asphalt mix we are talking?
24. 122-204 Lines: I don’t understand why you describe the test procedures when expert reading your paper knows the standard procedure of the tests and if doesn’t then may refer to standard and read it. You simply refer to the norm or standard and if there are differences in standards among countries then specify them. You don’t need to write in your review things known to readers specialized in this field.
25. 4.1. -> Any study from US about application of diatomite in asphalt considering that they have reserve of it?
26. 230 Line: 50% in comparison to what? No compaction problems?
27. 233 Line: amount of fibers used? You need to mention it.
28. In your chapter 4.1. I really miss key information: mix designs of the applied mixes, the percentage of used diatomite and fibers mentioned directly in the text (you have some info in Table 1 but no reference to it in the text, also it is not practical to read text and after check missing info in table, you need to polish your review text. Can you please rewrite this chapter and bring important details into your text?
29. 257 Line: To your notice addition PE doesn’t favor quality of the asphalt, now, why diatomite and PE are added together in the mix? At which conditions? Type of the mix? And both of the added additives work on the properties of the asphalt mix?
30. 262 Line: how high? It is always better to give values or %.
31. 275-276 Lines: You need to remove it from here and put before you start discussion about diatomite. Besides when you write a review, you need to make a thorough search of the publications trying to fill in your table as more as possible, that is a review.
32. 281-287 Lines: paragraph about what is lignin can be shortened and with one-two references, not 11!
33. 298 Line: asphalt mixes are mentioned for the first time, please check all your references and provide this info to all studies you mention in your review. And again, how many % of lignin fiber was used?
34. 308 Line: and the amount of CMR in the mix is….? Have you ever mixed yourself asphalt mix with lignin fibers? Any practical experience? What is optimal amount of lignin fiber can be added into mix?
35. 280-350 Lines: All this info can be simply placed in a very compact table with no repetition of the same phrases. I expect that in those lines you shortly write the outcome of each study and do not repeat info mentioned in table. Now it feels like info splitted into parts some info in text and some in table. When your review is read it should be clear what and how, and no look for missing info. You refer to the table in the beginning of the 4.2. not at the end in best case, or directly in the text where it mentioned for the first time. But then you will continuous number of repeats (see table 1, see table 1…), you need to think how better to reorganize it.
36. Tables 1 &2: you don’t need to write units in each table cell, just in table captions, for increased or decreased values you may use simply arrows (up/down).
37. 353-373 Lines: Provide extended version and detailed information (values and % for comparison always should be provided!) in your conclusions.
Hopefully my remarks and recommendations will be useful to improve the quality of your paper.
Regards,
Reviewer
Author Response
Reply to review comments
(Letters C & R denote comment and reply respectively)
We would like to thank you for your constructive comments, which certainly help us to improve the quality of our manuscript. As instructed and suggested, we have addressed all review comments on a point-by-point basis and made major revisions to our manuscript accordingly.
Reviewer #1:
C: My first critical comment will be about the list of references you have provided: you have 108 references (which is quite an amount) but only part of them refers to the application of lignin fiber and diatomite in the asphalt mix / pavement. Why? I don’t understand why you need to mention references with epoxy resin, gneiss, slag etc. Your main focus is at application of lignin and diatomite, so? You may decrease number of references and select only those which are really significant for the subject of your paper. But for example, to have 11 references in the paragraph for describing what is lignin is a bit too much too.
R: Done, we have decreased the number of citations from 108 to 85 citations.
We would like to illustrate that, this paper not only discussed the material which affect the asphalt mixes, but also discussed the pavement distresses caused by overloading on highways and how we can evaluate it by explaining the experimental laboratory tests.
C: Another critical comment is about the description of test methods, extraction of info from standard, why? usually experts reading your paper will need just a reference to standard mentioned in your paper. Therefore, I suggest to rewrite chapter 3.
R: Done, we have rewritten chapter 3 and improved it as mentioned from line (128-221).
We explained the test methods because previous studies follow the third chapter in terms of sample preparation, storage, and the number of tested samples. We mentioned it in order to increase the knowledge of readers and enable them to know where the results mentioned in the fourth chapter come from as well as the procedures of these tests.
C: Actually, I would suggest to reorganize your paper: start with short info on the problems of distresses in the flexible pavements in your intro; then you provide short info on the materials – diatomite and lignin fiber; then you write critical review about application of those additives in asphalt mixes with sub-chapters mentioned in chapters 2 and 3, in that way chapter 4 will be fully covered. Also, you need to write recommendations to the readers, preferably if you have practical experience mixing those additives in asphalt mix and your knowledge is not only based on the publications you have read. Also to provide comparison with other additives (here you may use the references about other materials but not in test description part as you did previously).
R: Done, we have reorganized our paper as per your comments (C8, C9) by adding the main advantage of flexible pavement in comparison to concrete pavements and the percentage of flexible pavements from line (32-37), problem statement and solution from line (38-48), short introduction additives from line (58-60) and recommendations to the readers, how they can mix those additives in asphalt mix from line (129-139).
C: More details have to be added into text, in order when reading it would be clear: which mixes used, conditions, amounts of additives, obtained results and comparison of those results. Technological aspects are mentioned in the references you refer to? It is important, because asphalt material is quite sensitive to the additives and they immediately affect its compaction, etc. There are a lot of studies, but all of those studies have a practical application? Reading your paper I need to see what is the optimal amount of addition of lignin and diatomite, for which mixes, for which environment/region and maybe you may provide asphalt mix design those additives are used? In case I would like to try lignin or diatomite for production of asphalt mix, I prefer to read a study where I can find already all basic necessary information. I miss it in your review.
R: Done, we have added all data mentioned in the previous studies in table 1 and 3 in line 311 and 399, respectively. Types and amounts of additives, condition of tests (test temperature) and results were mentioned in table 2 and 4 in line 312 and 401, respectively.
C: In your conclusions it should be stated why lignin and diatomite should be used in asphalt mixes, what are advantages and disadvantages, optimal amount of those materials could be added into mix, what type of mix, etc.
R: Done, we have included the advantages and disadvantages of additives as mentioned from line (411-418), and the optimal amount of those additives could be added into mix from line (419-420).
C1: 4 Line: Josephine Musanyufu?(name, surname) Also you don’t need to add “1” next to names if all authors are from the same institution.
R1: Done, we corrected the names in line 4,5 and we removed the number next to names.
C2: 10-11 Lines: I would suggest rephrasing this sentence, it can be written in a shorter form, basically the idea is that traffic density and overloading is drastically increased in comparison to some years ago. Percentage?
R2: Done, we rephrased the sentence from line (10-11). We have also tried our best to find the exact percentage of the increase of traffic density from the previous researches but we cannot find it related to the whole world.
C3: 11 Line: Pavements “suffer”, please insert another term.
R3: Done, we corrected this sentence from line (11-12).
C4: 14 Line: impairements?
R4: Done in line 13.
C5: 18 Line: The review demonstrated? Indicated?
R5: Done in line 17.
C6: 19, 57, 62 Lines: replace “scholars” with “researchers”
R6: Done in line (18, 56 and 64).
C7: You need to mention in your abstract the percentages of optimal fiber amount could be applied in the asphalt mixtures, and if you mention high or low temperatures then you need also show the range. Low might be 0 or -30, high +30 or +70.
R7: Done, we have mentioned the percentages of optimal fiber amount of Diatomite and lignin fiber in line 20 and 22, respectively.
C8: 32-34 Lines: Experts reading your paper do not need definition of the flexible pavements, therefore, I would suggest to remove those lines.
R8: Done, we have removed those lines and rewritten the paragraph from line (32-37).
C9: 34-36 Lines: the percentage of flexible pavements applied worldwide? Main advantage in comparison to concrete pavements?
R9: Done, we have rewritten this paragraph from line (32-37).
C10: 37-39 Lines: swift? Where this sentence comes from? I suggest to rewrite it.
R10: Done in lines 38 and 39.
C11: 44 Line: please choose which term you use throughout your manuscript “mix” or “mixture”. I would suggest “mix”. But don’t use both.
R11: Done, we have corrected it in the whole paper by using only (mixes).
C12: 44-45 Lines: Increase in thickness of layer is not really a good solution, if you have crack at the bottom the thickness of layer doesn’t really matter at all. Please add reference straight away in this sentence. I also suggest to add several references from your list who mention that increase of layer thickness is a solution.
R12: The solution of increasing the thickness of the layer was derived from the reference (Arabani M, Mirabdolazimi SM, Sasani AR (2010).The effect of waste tire thread mesh on the dynamic behaviour of asphalt mixtures. J. Constr. Build. Mater., 24: 1060-1068). However, these solutions have been removed, as per your general comment to reorganize the introduction. We put problem and the direct solution which we want to discuss from line (38-48).
C13: 44-49 Lines: only two solutions?
R13: These two solutions were only mentioned as examples, though there are many solutions. But, we removed these solutions, as per your general comment to reorganize the introduction. We put problem and the direct solution which we want to discuss from line (38-48).
C14: 58 Line: Please add reference
R14: Done in Line 57.
C15: 68 Line: water effects? Specify please
R15: Done in Line 73 and we have also explained the water damage effects on the pavement in subchapter 2.3 from line (110-127).
C16: 84 Line: this study is focused on wheel tracking test… study or review?
R16: Done in line 89.
C17: 87 Line: I don’t really agree that cracking failures observed in the cold areas of the world. It is observed everywhere. Please indicate what type of cracking you mention in this line. Combine it with lines 92-93.
R17: Done, we have corrected this sentence in line 92 and 93. The type of crack mentioned is thermal cracking and have corrected in line 97.
C18: 88 Line: manifested? Another term please.
R18: Done, we have changed it in line 93.
C19: 119 Line: define/list which methods?
R19: Done from line (125-126).
C20: 121 Line: change to “freeze-thaw splitting”
R20: Done, we have changed it in line 127.
C21: 128-132 Lines: Strange switch, I think you need to start sentence not with “before the test” but in “According to standard [73]…”
R21: Done, we have changed it in line 147.
C22: I would suggest to rewrite/add more info 3.1.1. it doesn’t contain sufficient info for the review. Please in all your references in the list of references, mention all names of all co-authors to the publications you refer. It can be mentioned in the text, for example, Moreno et al. but in list of references the reference should contain all names contributed for that particular study.
R22: Done, more information has been added in this section from line (149-152) and all references in the list have been corrected.
C23: 130 Line: 60 degrees? The temperature of the mix? What type of asphalt mix we are talking?
R23: Done, 60oC is the room temperature and we have corrected it in line 149. The type of asphalt mixes was mentioned in the table (1 and 3) in line (311 and 399, respectively).
C24: 122-204 Lines: I don’t understand why you describe the test procedures when expert reading your paper knows the standard procedure of the tests and if doesn’t then may refer to standard and read it. You simply refer to the norm or standard and if there are differences in standards among countries then specify them. You don’t need to write in your review things known to readers specialized in this field.
R24: Done, we have rewritten chapter 3 and improved it as mentioned from line (128-221). We explained the test methods because previous studies follow the third chapter in terms of sample preparation, storage, and the number of tested samples. We mentioned it in order to increase the knowledge of readers and enable them to know where the results mentioned in the fourth chapter come from as well as the procedures of these tests.
C25: 4.1. -> Any study from US about application of diatomite in asphalt considering that they have reserve of it?
R25: Done, the information from the reference has been added in line 229.
C26: 230 Line: 50% in comparison to what? No compaction problems?
R26: Done, 50% of lime stone powder (8.1% of lime stone powder and 6.5 % of diatomite) of total mass of aggregate and we added it in line 261 and 262. No compaction problem was mentioned in this study.
C27: 233 Line: amount of fibers used? You need to mention it.
R27: Done, the amount of diatomite and basalt fiber has been mentioned in table 2 in line 312 with (Ref. number 17).
C28: In your chapter 4.1. I really miss key information: mix designs of the applied mixes, the percentage of used diatomite and fibers mentioned directly in the text (you have some info in Table 1 but no reference to it in the text, also it is not practical to read text and after check missing info in table, you need to polish your review text. Can you please rewrite this chapter and bring important details into your text?
R28: Done, we have added all data mentioned in the previous studies in table 1 and 3 in line 311 and 399, respectively. Types and amounts of additives, condition of tests (test temperature) and results have been mentioned in the table 2 and 4 in line 312 and 401, respectively. All data in the tables have been connected with the text by using the reference number which has been mentioned in the table and in the test and also we have added a paragraph from line (247-250) to connect the data in tables and texts. The data mentioned in the table was not mentioned again in the text to avoid repetition of the same data and phrases as you advised me in the comment (C35).
C29: 257 Line: To your notice addition PE doesn’t favor quality of the asphalt, now, why diatomite and PE are added together in the mix? At which conditions? Type of the mix? And both of the added additives work on the properties of the asphalt mix?
R29: Done, this type of the mix was mentioned in table 1 in line 311 with (Ref. number 21). The amount of PE was not mentioned in this literature. Diatomite and PE are added together because diatomite has a slight effect on the low temperature properties of asphalt mixes, but the PE has a significant influence on the low temperature crack resistance as added from line (292-293).
C30: 262 Line: how high? It is always better to give values or %.
R30: Done, we have corrected the test name in line 294.
C31: 275-276 Lines: You need to remove it from here and put before you start discussion about diatomite. Besides when you write a review, you need to make a thorough search of the publications trying to fill in your table as more as possible, that is a review.
R31: Done, this has been removed and placed from line (242-246). Diatomite was used in modified asphalt binder and modified asphalt mixes. But our study concentrated only on the Diatomite modified asphalt mixes, which we have mentioned.
C32: 281-287 Lines: paragraph about what is lignin can be shortened and with one-two references, not 11!
R32: Done, we have rewritten it from line (318-326) and decreased the number of references to three.
C33: 298 Line: asphalt mixes are mentioned for the first time, please check all your references and provide this info to all studies you mention in your review. And again, how many % of lignin fiber was used?
R33: Done, we have corrected it in the whole paper by using only (mixes). The amount of fibers is already mentioned in table 4 in line 401.
C34: 308 Line: and the amount of CMR in the mix is….? Have you ever mixed yourself asphalt mix with lignin fibers? Any practical experience? What is optimal amount of lignin fiber can be added into mix?
R34: Done, the amount of CRM has already been mentioned in table 4 in line 401 with (Ref. number 14)
C35: 280-350 Lines: All this info can be simply placed in a very compact table with no repetition of the same phrases. I expect that in those lines you shortly write the outcome of each study and do not repeat info mentioned in table. Now it feels like info splitted into parts some info in text and some in table. When your review is read it should be clear what and how, and no look for missing info. You refer to the table in the beginning of the 4.2. not at the end in best case, or directly in the text where it mentioned for the first time. But then you will continuous number of repeats (see table 1, see table 1…), you need to think how better to reorganize it.
R35: Done, We have reorganized the tables and added all data mentioned in the previous studies in table 1 and 3 in line 311 and 399, respectively. Types and amounts of additives, condition of tests (test temperature) and results have been mentioned in the table 2 and 4 in line 312 and 401, respectively. All data in the tables were connected with the text by using the reference number mentioned in the table and in the test and we have also added a paragraph from line (336-339) to connect the data in tables and texts.
C36: Tables 1 &2: you don’t need to write units in each table cell, just in table captions, for increased or decreased values you may use simply arrows (up/down).
R36: Done, we have reorganized the tables 1 and 3 in line 311 and 399, respectively.
C37: 353-373 Lines: Provide extended version and detailed information (values and % for comparison always should be provided!) in your conclusions.
R37: Done, we have mentioned the optimal amount of those additives, which could be added into mix from line (419-420).
Reviewer 2 Report
The article presents a review of using two different additives (Diatomite and Lignin Fibers) as a bitumen modification for usage in asphalt mixtures. The review describes the performance of mixtures with mentioned additives in whole range of temperatures using laboratory test commonly used in China. While the whole article seems to be written properly, it needs strong improvement and any kind of comment of found information. Detailed remarks are presented below:
1. The manuscript title suggest that the article is a review of the performance of two different additives used in asphalt mixtures, used literature do not correspond to the title. From 108 citations only around 20 are about the performance. Additional few describes basic properties of the material. This seems not right. The rest of the literature is used for general introduction and about the description of the test methods and their general results. The proportion should be opposite!! It should be review about the materials not about test methods and their validation
2. On what basis those two additions were selected? From the manuscript it is not clearly stated. There is no description how those two additions interact with each other? Whether they can be used in the same time? Please give clear motivation of the review.
3. Nevertheless the section about test methods needs improvement: please add detailed description of the sample preparation (compaction methods and procedure, used equipment). In some cases general description are added like: “ compacted 75 times each side” /with what equipment/, “compacted with a steel roller for 24 times according to Marshal test” /what information are from Marshal test? /. Please give detailed and clear description.
4. Add related standard specification for each test. Every presented test is strictly normalized in the whole world.
5. Line 2 – Surface course is not called Asphalt Concrete (AC). AC is one of the materials used for surface course. Please correct the statement.
6. Section 4 and 5 needs strong improvement.
a. Please provide all information regarding used materials: (1) asphalt mixture type (asphalt concrete, stone mastic asphalt, sand asphalt, etc), (2) composition of the used asphalt mixture (type and amount of bitumen, type and amount of aggregate, basic aggregate properties, voids), (3) description of the used materials – type of bitumen, aggregate, fillers, (4) amount of the used additive – is it related to the bitumen or the whole mixture (5) role of the used additive – modification / filler replacement
b. Information regarding specimen preparation and storage. Was ageing applied to bitumen or the mixture?
c. How many specimen were investigated?
d. Tables with abbreviations needs proper description.
7. Described research results should be thoroughly commented. As stated before, for example there is no information regarding the interaction between additives. Only general conclusion are presented.
8. Some editorial mistakes should be corrected (i.e. o as a degree mark “5oC”)
Author Response
Reply to review comments
(Letters C & R denote comment and reply respectively)
We would like to thank you for your constructive comments, which certainly help us to improve the quality of our manuscript. As instructed and suggested, we have addressed all review comments on a point-by-point basis and made major revisions to our manuscript accordingly.
Reviewer #2:
C1: The manuscript title suggest that the article is a review of the performance of two different additives used in asphalt mixtures, used literature do not correspond to the title. From 108 citations only around 20 are about the performance. Additional few describes basic properties of the material. This seems not right. The rest of the literature is used for general introduction and about the description of the test methods and their general results. The proportion should be opposite!! It should be review about the materials not about test methods and their validation.
R1: Done, we have decreased the number of citations as first reviewer advises us from 108 to 85 citations.
We would like to illustrate that, this paper not only discussed the material which affect the asphalt mixes, but also discussed the pavement distresses caused by overloading on highways and how we can evaluate it by explaining the experimental laboratory tests.
C2: On what basis those two additions were selected? From the manuscript it is not clearly stated. There is no description how those two additions interact with each other? Whether they can be used in the same time? Please give clear motivation of the review.
R2: Done, the paragraph from line (61-68) has been reorganized and concerning the use of these two additives together in the same mixes, no one has done it before. It’s recommended to be a future study.
C3: Nevertheless the section about test methods needs improvement: please add detailed description of the sample preparation (compaction methods and procedure, used equipment). In some cases general description are added like: “compacted 75 times each side” /with what equipment/, “compacted with a steel roller for 24 times according to Marshal Test” /what information are from Marshal test? /. Please give detailed and clear description.
R3: Done, this has been rewritten from line (128-221), where we have mentioned the preparation of samples and storage, number of specimens required for each test, number of compaction and compaction equipment.
C4: Add related standard specification for each test. Every presented test is strictly normalized in the whole world.
R4: Done in line (147, 166, 184, 204 and 217).
C5: Line 2 – Surface course is not called Asphalt Concrete (AC). AC is one of the materials used for surface course. Please correct the statement.
R5: Done, we have removed this line, as the first reviewer asks us to remove the first paragraph and the new paragraph has been written from line (32-37).
C6.a: Please provide all information regarding used materials: (1) asphalt mixture type (asphalt concrete, stone mastic asphalt, sand asphalt, etc.), (2) composition of the used asphalt mixture (type and amount of bitumen, type and amount of aggregate, basic aggregate properties, voids), (3) description of the used materials – type of bitumen, aggregate, fillers, (4) amount of the used additive – is it related to the bitumen or the whole mixture (5) role of the used additive – modification / filler replacement
R6.a: Done, we have added all data mentioned in the previous studies in table 1 and 3 in line 311 and 399, respectively and the amounts of additives have been mentioned in table 2 and 4 in line 312 and 401, respectively.
C6.b: Information regarding specimen preparation and storage. Was ageing applied to bitumen or the mixture?
R6.b: Done, it has been mentioned in the third chapter from line (128-221), where all previous studies follow the third chapter in terms of sample preparation, storage, and the number of tested samples.
C6.c: How many specimens were investigated?
R6.c: Done, it has been mentioned in the third chapter from line (128-221), where all previous studies follow the third chapter in terms of sample preparation, storage, and the number of tested samples.
C6.d: Tables with abbreviations needs proper description.
R6.d: Done in line 431
C7: Described research results should be thoroughly commented. As stated before, for example there is no information regarding the interaction between additives. Only general conclusions are presented.
R7: Done, we have mentioned the significant reason for the effect of additives on the performance of asphalt mixes from line (307-310) and (394-398) for diatomite and lignin fiber, respectively.
C8: Some editorial mistakes should be corrected (i.e. o as a degree mark “5oC”)
R8: Done, we have checked all degree mark in the manuscript.
Reviewer 3 Report
1) It would be useful to have a few pictures of Diatomite and Lignin fibers in the literature review. Authors can take permission from the previously published papers' publishers/editors to put the images in the literature review.
2) What are the physical properties such as density, surface texture, and gradation of Diatomite and Lignin fibers? Please provide numerical values.
3) According to line 217, it says Diatomite can be used to reduce environmental pluution, but according to line 213, it says Diatomite is non-toxic. How does it pollute environment if it is not-toxic?
4) What is the authors' recommendation on possible mix design using both Diatomite and Lignin fibers based on the literature review?
5) What is the authors' recommendation on the improvement of mix design using both Diatomite and Lignin fibers based on the literature review?
6) What can other laboratory tests be done to understand the effects of Diatomite and Lignin in the asphalt concrete mix?
Author Response
Reply to review comments
(Letters C & R denote comment and reply respectively)
We would like to thank you for your constructive comments, which certainly help us to improve the quality of our manuscript. As instructed and suggested, we have addressed all review comments on a point-by-point basis and made major revisions to our manuscript accordingly.
Reviewer #3:
C1: It would be useful to have a few pictures of Diatomite and Lignin fibers in the literature review. Authors can take permission from the previously published papers' publishers/editors to put the images in the literature review?
R1: This has been done in line (239) and (328).
C2: What are the physical properties such as density, surface texture, and gradation of Diatomite and Lignin fibers? Please provide numerical values.
R2: This has been done from line (234-233) and (325).
C3: According to line 217, it says Diatomite can be used to reduce environmental pollution, but according to line 213, it says Diatomite is non-toxic. How does it pollute environment if it is not-toxic?
R3: Diatomite can absorb slight content of bitumen. By this way, the total amount of bitumen in asphalt mixes will be reduced. So, Diatomite can decrease the total pollution in the environment, which is caused by bitumen.
C4: What is the authors' recommendation on possible mix design using both Diatomite and Lignin fibers based on the literature review?
R4: Done, this has been added in subchapter (3.1.) from line (129-139).
a. Diatomite and bitumen are weighed to the required amount, and left in the oven at 135OC for 4 h. The diatomite and the bitumen are then taken out of the oven and mixed together with the blending equipment at a speed of 600 r/min for 15 min.
b. Lignin fibers are mixed with the hot aggregates for a minimum of 30 s to enhance fiber dispersion; then the diatomite modified asphalt is added while continuing to mix for nearly 2 additional minutes. The overall processing time should not exceed 6 min in order to prevent asphalt aging.
C5: What is the authors' recommendation on the improvement of mix design using both Diatomite and Lignin fibers based on the literature review?
R5: Asphalt mixes modified with two varieties of admixtures (Diatomite and lignin fiber) can be utilized to improve High temperature stability, Low temperature stability and Water stability of asphalt mixes.
C6: What can other laboratory tests be done to understand the effects of Diatomite and Lignin in the asphalt concrete mix?
R6: a. Four point bending beam test (Fatigue life test).
b. Uniaxial Compression Repeated Creep-Recovery Test (Creep test).
Round 2
Reviewer 1 Report
Dear Authors,
thank you for your revised manuscript, it is obvious that some work has been done within last 10 days and you have improved the quality of your Review. I have read it once again, in my opinion it looks better now, but I still have some comments to your manuscript and it would be advisable for You to read it once again very carefully and bring updates (please pay attention to technical terminology and try to choose one term for all the manuscript (mix or mixtures, specimens or samples, etc.).
10-11: should be corrected to -> Due to rapid growth of traffic density, the phenomenon of overloading on high-grade highway causes various modes of distresses to the pavement such as: rutting,... (It would be not correct to mention asphalt mixes here, the modes of distesses are related to the pavement).
12: ...is the most common solution (not popular)...
13: should be corrected to -> the performance of asphalt pavement to mitigate its damages.
15: many tests -> ...several tests, such as: wheel tracking, indirect tensile, three point bending, freeze thaw splitting and marshall immersion were reviewed.
20, also 419: ... is 12~14% of asphalt binder can be added into mix.
22-23, also 420: ...0.2-0.4% per asphalt mix composition. (to write “of asphalt mixes” is not precise)
23: The review also indicated... (or within the review it has been observed)
24: I don't agree that you can name lignin fiber and diatomite as admixtures. Both are additives. It is always useful to check technical vocabulary while writing.
27: review predicts? Doesn't really sound. Better use “review suggests”
61: Previously, researchers noted that diatomite has a key role....
64: ...has minor effect (not little)
65: ...so it can be verified... (it is better to skip while writing “we, me, I, he etc.”)
68: asphalt mix.
145-152: according to standard ... three slab specimens (dimensios?) are prepared... (not “were” prepared, it is usual procedure then you use verb in present). Test samples or specimens? Please use one term for it. The same for lines 162-171.
196, 214: samples or specimens?
247-250: Grammar item, you need to rephrase your lines in order to a better comprehension.
252: mix (not Mix)
258: low (not Low) temperature (how many degrees?)
307, 311, 425: diatomite (not Diatomite, why from capital letter?)
313, 402: why do you add % unit in the 3rd column in tables? you have already indicated % in the table legend.
317: Properties of lignin fiber
394: aforementioned? -> from above mentioned studies, it can be concluded...
431-432: why do you need to write abbreviations separately, you have mentioned all in the text?
Regards,
Reviewer
Author Response
The minor revisions have been highlighted with Turquoise color in the manuscript.
Reviewer 2 Report
Thank you very much for implementing all of the remarks.
The presented article is much better than presented in previous form.
All necessery information regarding specimen, laboratory tests and their results have been added.
All standard specification have been added.
Vague statements were either corrected or removed.
Author Response
We would like to thank you for your reply, interest, support and constructive comments, which certainly help us to improve the quality of our manuscript.
Round 3
Reviewer 1 Report
Dear Authors,
thank you, your manuscript looks very good now, I believe the final version after proof check will be even better. I think Editor will decide whether abbreviations should be written separately or not, in my opinion it is sufficient when it is mentioned in the text. Good luck with your research!
Regards,
Reviewer